# LoRaCog: A Protocol for Cognitive Radio-Based LoRa Network

**DOI:** 10.3390/s22103885

**Published:** 2022-05-20

**Authors:** Firas Salika, Abbass Nasser, Maxime Mroue, Benoît Parrein, Ali Mansour

**Affiliations:** 1Intelligent Computing and Communication Systems Lab, Computer Science Department, American University of Culture and Education, Beirut 1507, Lebanon; fms014@auceonline.com (F.S.); abbassnasser@auce.edu.lb (A.N.); 2Laboratoire des Sciences et Technologies de l’Information, de la Communication et de la Connaissance, ENSTA Bretagne, 29806 Brest, France; 3Syndicat d’Énergie Intercommunale de Maine et Loire, 49000 Ecouflant, France; husseinmrouehh@gmail.com; 4Nantes Université, LS2N UMR CNRS 6004, 44306 Nantes, France; benoit.parrein@univ-nantes.fr

**Keywords:** cognitive radio, IoT, LPWAN, LoRaWAN, LoRaCog, licensed spectrum, unlicensed spectrum, rejected packet rate, primary users, secondary users

## Abstract

In this paper, we propose a new protocol called LoRaCog to introduce the concept of Cognitive Radio (CR) in the LoRa network. LoRaCog will enable access to a wider spectrum than that of LoRaWAN by using the unutilized spectrum and thus has better efficiency without impacting the end devices’ battery consumption. LoRa networks are managed by LoRaWAN protocol and operate on the unlicensed Industrial, Scientific and Medical (ISM) band. LoRaWAN is one of thriving protocols for Low-Power Wide-Area Networks (LPWAN) implemented for the Internet of Things (IoT). With the growing demand for IoT, the unlicensed spectrum is expected to be congested, unlike the licensed spectrum, which is not fully utilized. This can be fairly balanced by applying CR to the LoRa network, where the End Devices (EDs) may change the operating channel opportunistically over the free/available licensed spectrum. Spectrum sensing, channel selection and channel availability relevance become essential features to be respected by the proposed protocol. The main objective of adding CR to LoRaWAN is reducing the congestion and maintaining LoRaWAN’s suitability for battery-operated devices. This is achieved by modifying LoRaWAN components such as the ED receive window RX2 rearrangement, spectrum sensing functionality by gateway (GW) for identifying unused channels, and reaching a decision on the unused channels by network server (NS). These changes will create LoRaCog meeting spectrum efficiency and maintain the same level of battery consumption as in LoRaWAN. Numerical simulations show a significant decrease in the rejected packet rate (more than 50%) with LoRaCog when more EDs use cognitive channels. As the results prove, LoRaWAN can reach above 50% rejected packets for the simulated environment versus 24% rejection for LoRaCog using only one additional channel (means total two channels). This means that the system can eliminate rejected packets almost completely when operating over the possible many channels. As well, these results show the flexibility in the system to utilize the available frequencies in an efficient and fair way. The results also reveal that a lower number of GWs is needed for LoRaCog from LoRaWAN to cover the same area.

## 1. Introduction

Statistics foresee 50 billion IoT objects to be connected by 2030 [1]. Wireless communications including IoT use different protocols that vary between proprietary to interoperable ones. LoRaWAN is an interoperable IoT wireless communication protocol used for Long-Range (LoRa) networks. Its system architecture ensures a long range and a low-power consumption at a low bit rate. The battery lifetime, network capacity, Quality of Service (QoS) and security enable LoRaWAN to be used in various applications [2]. Several applications are integrating LoRaWAN protocol, especially those involving sensors such as smart metering, smart parking, smart street lighting, vehicle fleet tracking [3], in agriculture such as smart farming [4,5], and in environment or atmospheric monitoring [6].

The majority of the IoT wireless networks operate on the free unlicensed spectrum. The ever-growing demand for the IoT technologies will increase the load on the burdened unlicensed spectrum. Therefore, the Cognitive Radio (CR) technique can address such a challenge by sharing unused licensed spectra. CR shares the spectrum between two types of users: primary and secondary users. Primary users (PU) are allowed to access the licensed spectrum whenever needed, while the secondary users (SU) can transmit on the licensed spectrum only when the primary users are idle or the channel is not in use. To manage this sharing process, the secondary users are responsible for making spectrum sensing before transmitting to avoid any interference with the primary users. In LoRaCog, when applying CR, the IoT devices are acting as secondary users.

As it is known, one of the CR’s main challenges is the high power consumption due to spectrum sensing [7], while on the other hand, the main advantage of LoRaWAN is its suitability for battery-operated devices. This forms the relationship and incentive for combining both technologies, leading to the delivery of LoRaCog. Usually, the end devices (EDs) in LoRaWAN are battery powered, making CR implementation a real challenge. On the other hand, CR and LoRaWAN are partially similar in having multiple channel options; for example, LoRaWAN ED can use one of several channels within the allowed frequency. Moreover, as per LoRaWAN protocol, EDs are provided by the channel’s parameters for their next uplink. These characteristics make the CR technique, where the channel allocation is dynamic, suitable to LoRaWAN specifications after undergoing certain modifications. Indeed, CR means operation on the licensed bands sometimes, which could result in interference with the existing users. However, thanks to the spectrum sensing mechanism performed by the CR-based LoRa network, this interference can be minimized to a tolerable threshold.

In our recent work [8], we adapted LoRaWAN protocol to cope with spectrum sensing. The modification applied to LoRaWAN allows the EDs to sense on demand a given channel and provide a CR-based network with the status of that channel, while the EDs themselves operate only on the unlicensed bands.

This research presents the new protocol “LoRaCog” that allows the LoRa network to dynamically operate on both the licensed and the unlicensed bands using CR. We exploit the rearrangement of the preamble frame in LoRa suggested by our previous work LoRa+ [9], and we suggest a new mechanism for spectrum sensing by Gateway (GW) to identify the unused spectrum and convey the parameters (unused spectrum) to EDs based on the decision by the Network Server (NS). These modifications result in operating the LoRa network over both the available licensed and the unlicensed spectrum alleviating the congestion on the unlicensed spectrum and decreasing the rejected packets rate. It is worth mentioning that LoRaCog does not replace LoRaWAN, but it can be considered as a complementary solution that assists LoRaWAN to avoid the high congestion on the unlicensed bands and open the door to use the licensed bands using CR technology. LoRaWAN remains adopted when the EDs operate on the unlicensed spectrum, while LoRaCog extends the access for LoRa transmissions to the licensed frequencies.

The main contribution of the paper can be listed as follows:A new LPWAN protocol that provides access to the unutilized or booked spectrum without impacting the primary user’s performance.LoRaWAN protocol modification to adapt the cognitive radio requirements maintaining the same battery consumption of LoRaWAN EDs by allocating spectrum sensing to GWs.Rearrangement of the ED’s receive window giving the chance to collect the broadcasted unused channels by GWs.NS new role to receive all sensing from GWs and allocate the channels that suit each ED by correlating the best RSSI.Extensive simulations are performed to verify the efficiency of the proposed protocol simulating real scenarios from different distribution of EDs and GWs to the appearance of primary users.

The rest of the paper is organized as follows: Section 2 presents recent works on the performance of LoRaWAN and its cognitive implementation. Section 3 shows a brief explanation and highlights the CR concept, which is followed by Section 4, where the current LoRaWAN standard communication protocol is presented. The suggested new protocol and modifications on the current LoRaWAN protocol are detailed in Section 5. Performance evaluation and simulation environment are presented in Section 6. Then, results are discussed in Section 7. Finally, Section 8 summarizes and concludes our work.

## 2. Related Works

CR and IoT domains have been actively researched in the last decades. Only a few researchers spoke about CR with a low-power wide-area network (LPWAN or LPWA network) but remained generic about cognitive IoT.

Moy and Besson [10] proposed a solution to mitigate radio collisions in IoT unlicensed bands. It is based on learning algorithms implemented on the IoT device side, at a very low cost of implementation and no protocol overhead. The work was demonstrated on LoRa devices in a real LoRaWAN network and named IoTligent. The results show that the system is able to use the available channels by learning on the spot when ISM sub-bands are less occupied or jammed than others. Adelantado et al. [11] discuss the limitations of LoRaWAN and raised an open question about the CR. From the author’s point of view, the inclusion of CR into the LoRaWAN standard could significantly reduce the energy consumption. In [12], the authors proposed an adaptable protocol for LoRa networks with low overhead and complexity. The work introduces an adaptive algorithm for LoRa in two different frequencies. The results obtained by a real scenario show an SNR improvement of 4.68%. Chen et al. [13] introduce a cognitive low-power wide-area network (Cognitive-LPWAN) architecture to safeguard stable and efficient communications in a heterogeneous IoT. The work proposes the usage of artificial intelligence (AI) to balance the demand between heterogeneous IoT devices with communication delays and energy consumption. The experimental results show that this scheme can meet the communication’s demand in applications where appropriate communication technologies are chosen to achieve a better interaction experience. Marquet et al. [14] presented different LoRa demodulation and decoding strategies, Software-Defined Radio (SDR) implementations, as well as performance assessment under various channel conditions. The simulations show that LoRa systems have good properties for time and/or frequency selective channels, thanks to the robustness of its underlying Chirp-Spread Spectrum (CSS) modulation. Although CR is a promising solution to solve spectrum inefficiency, it brings challenges especially on the hardware design and cost levels as highlighted in [15] or at the complexity of managing such networks in LPWAN systems where the signal is going to propagate for long distances that add to hardware and software challenges, which is a security concern that will require heavy encryption techniques and thus more energy consumption, as stated in [16].

Other researchers worked on LoRaWAN performance optimization without considering CR, such as Zorbas et al. [17], who showed that a GW can maximize the network capacity by optimizing the spreading factor (SF) distribution to EDs. The work presents combinations of bandwidths and spreading factors to enhance the capacity of LoRa networks and compute the packet success ratio in each combination. Reynders et al. [18] suggest an optimization algorithm of SF and transmission power while avoiding near–far perturbations. This is achieved by allocating distant users to different channels. The near–far perturbations mean that a packet of an ED far from the gateway suffers destructive collisions more than a packet of an ED close to the GW. Simulations show that the packet error rate can be decreased down to 50% for distant EDs in a moderate contention scenario. Cuomo et al. [19] propose EXPLoRa, which improves the data rate and the network’s robustness compared to the adaptive data rate (ADR) mechanism. It allocates proper SFs to EDs based on distance, Received Signal Strength Indicator (RSSI), and the number of connected EDs in the network. Croce et al. [20] evaluated collisions in high-density LoRa networks. The work shows imperfect orthogonality due to different SFs. Larger SFs in congested networks are most likely supposed to have collisions. Mroue et al. [9] suggested LoRa+, a new mechanism that improves the QoS of LoRaWAN. The work explains LoRaWAN inconvenience when transmitting on a channel before receiving the parameters from the GW. This drawback increases the rejected packets rate within the LoRaWAN network and decreases the radio coverage of the GW. LoRa+ is based on the LoRaWAN protocol with modifications at the RX2 slot, which will be opened just before the uplink transmission (Tx frame). The results show a significant decrease in the collision and decrease the number of GWs by 80% within a high-density area of ED.

On the other side, IEEE 802.22 is a CR-based standard for a wireless regional network using white spaces in the television (VHF/UHF TV broadcast bands) frequency spectrum between 54 and 862 MHz, potentially at the 1300 to 1750 MHz and 2700 to 3700 MHz where regulations allow [21]. This development is aimed at using CR techniques to allow the sharing of a geographically unused spectrum allocated for TV (white space) without causing interference. The standard specifies the air interface, including the cognitive medium access control layer (MAC) and physical layer (PHY). Sub-GHz such as the UHF bands are attractive for the LoRa implementation thanks to the ability to cover wide areas with a low power. Furthermore, 5G will open new frequency bands in the sub-GHz spectrum [22]. These bands might be accessed by the LoRa devices acting as secondary users when CR is applied.

The presented state-of-the-art works analyzed scenarios in which LoRa performance is generally well studied and detailed when cognitive concept is applied directly or indirectly. The most relevant work is what have seen in [9,13], but our work in LoRaCog generates a fully operational protocol that allows the LoRa network to operate on both licensed and unlicensed bands, and it can be used as a complementary one for LoRaWAN when needed. We detail in this work the CR tasks assigned to the LoRa networks elements, i.e., spectrum sensing, spectrum decision, and the resulting action. By exploiting the geo-location of the GWs and their spatial distribution, they are responsible for performing a cooperative spectrum sensing and reporting the results to the NS. The latter is responsible for making the final decision on the channel availability based on the received sensing from the GWs and the historical data it stored on the cognitive operating channels. Once the decision on a given channel is taken, the EDs will be informed accordingly to operate on these channels. In addition, the proposed protocol is adaptable to a heterogeneous use of licensed/unlicensed bands by exploiting the diversity of the SFs. This feature facilitates the adoption of LoRaCog to operate simultaneously in parallel with LoRaWAN.

## 3. Cognitive Radio Overview

The need for wireless communication has been increasing since its existence. Most devices are being transferred from conventional wired into smart and wireless to cope with technology expansion and life needs. Not only that, but wireless communication opened the doors for innovations that become essential for daily life usage involving advanced telecommunications use cases, devices, sensors, equipment, etc. Such evolution increased the demand for radio frequencies, especially within the unlicensed spectrum. On the other hand, the licensed spectrum is underutilized.

The spectrum worldwide can be classified into categories depending on each country’s regulations. For simplicity, we will categorize globally into three: Used-Licensed, Used-Unlicensed, and Reserved-Not Used. Eventually, the radio spectrum can experience either congestion or inefficient utilization depending on many factors, but the entire spectrum will never be efficiently used. Luckily, researchers anticipated early for this problem to find a promising technique to overcome such challenges known as CR. The concept was first proposed by Joseph Mitola in 1998 and published late in 1999 [23].

CR is a self-learning system that detects and analyzes its environment, adapts to changes, and acts to use the radio spectrum efficiently. CR scans the spectrum for licensed users recognition to allow unlicensed users to utilize the assigned spectrum when it is not used as illustrated in Figure 1 [24]. Spectrum holes are the channels that are available and not in usage at the time and space of sensing [25,26].

The CR mechanism consists of three cycles starting by spectrum sensing, deciding, and then acting [28]. The cycle starts with a critical process where the spectrum sensing is performed. This can be up to a certain accuracy depending on the environment and other factors such as sensing technique, possible multipath fading, shadowing, or varying channel conditions [29].

In the next process, SUs make a decision based on what has already been observed using their knowledge, which may have been impacted by the uncertainty in the detected measurements, leading to imperfect decision on the channel availability.

In the last process, the CR device will apply the decision concluded from the learned information, i.e., choose the channel to transmit on and avoid the channels that are diagnosed as occupied. The action may depend on the uncertainty of the cycle’s steps.

Despite its promising objective, CR has been facing implementation challenges, especially for battery-powered devices due to the need for continuous channel monitoring. The more precision needed for spectrum sensing, the more sensing time is required, which has an impact on battery lifetime for battery-powered devices [30,31].

In addition, sensing accuracy may be affected by the sensing method itself and the channel conditions between PU and SU, such as the fading and shadowing. Moreover, the noise uncertainty may degrade the sensing accuracy, leading to misdetecting the true PU status or decreasing the spectral efficiency of SU [15,29].

The performance of the CR system may also be impacted by the hardware design. Most of today’s radio devices are compact and portable, being cost-effective modules, especially those working under unlicensed spectrum, and that obliges designers to use antennas with limited performance [32,33].

Hence, the successful identification of spectrum opportunities depends on the efficiency of the sensing technique as well as the unlicensed user specification. Hence, CR will be easily implemented with radio protocols that have similar specifications in nature. That will reduce the workload and investment. LoRaWAN protocol can adapt CR due to flexibility when it comes to changing the frequency within limited bandwidth and regulations.

## 4. LoRaWAN Protocol

Smart cities, utilities, companies with fleets on the ground (transport, maintenance, delivery, etc.), agriculture, health and many other applications are thriving for radio protocols that help them better manage and control their work as well as provide better service for users [34,35,36].

IoT is connecting objects everywhere, where LPWAN is a type of wireless telecommunication protocol for IoT. LPWAN characteristics such as long range, low power, low cost and multi-usage are the main features. Many mature networks such as LoRa [37], Sigfox [38], Wi-SUN [39] and many others were proposed as LPWAN solutions to meet the desired targets.

LoRa is an LPWAN technology developed by Cycleo and then acquired by Semtech [40]. It is based on spread spectrum modulation techniques derived from the Chirp Spread Spectrum (CSS) [41]. LoRaWAN defines the communication protocol and system architecture for the network, while the LoRa physical layer enables a long-range communication link. LoRaWAN protocol and its network architecture are main contributors in preserving the node’s battery lifetime, network scalability, security, and the variety of applications served by the network.

### 4.1. LoRaWAN Regional Parameters and Classes

This section discusses the main components that will undergo modifications to form LoRaCog. It is important for our research’s objective to give an overview of these components. LoRa utilizes unlicensed ISM bands, i.e., 868 MHz in Europe, 915 MHz in North America, and 433 MHz in Asia.

To easily follow the work later in this paper, we highlight here LoRa’s regional parameters for EU bands (European Union) [42]. The protocol requests for three channels to be set at manufacturing by default for every ED (868.1, 868.3 and 868.5 MHz with a bandwidth (BW) 125 kHz), and the rest of the channels will be dynamically allocated during operation. Those channels are the minimum set that all network gateways shall be listening to and cannot be modified through the “NewChannelReq” command to guarantee a minimal common channel set between EDs and network gateways.

LoRaWAN EDs can implement different classes as shown in Figure 2. LoRa is a bidirectional communication protocol, which means communication between ED and GW can flow in both directions. As detailed in the LoRaWAN 1.0.3 specification document [42], there are three classes (Class A, B and C). The EDs of Class A (see Figure 3) allow for bidirectional communications whereby each uplink transmission is followed by two short downlink receipt windows. The transmission slot scheduled by the ED is based on its own communication needs with a small variation based on a random time basis (ALOHA-type of protocol) [43].

The Class A operation is the lowest power system for applications that only require downlink communication from the server shortly after the ED has sent an uplink transmission. Downlink communications from the server at any other time will have to wait until the next scheduled uplink.

LoRaWAN works as star topology, which means GW is in a direct link with EDs. In other words, the EDs can communicate only with GW and not with each other.

The system consists, in addition to EDs of GWs, a Network Server (NS), a Join Server (JS), and an application server. As the intention is not to explain LoRaWAN in detail, we will explain some main elements that contribute and/or undergo modifications to produce LoRaCog in the following section.

### 4.2. LoRaWAN Key Features

The objective of this section is to explain the major specifications of LoRaWAN mentioned in the introduction, which makes it a successful candidate to adopt CR.

#### 4.2.1. Spreading Factors (SF)

LoRa is a Chirp-Spreading Spectrum (CSS) technology that offers good resistance to interference. The long communication distance is obtained at the expense of the transfer rate. The LoRa technology is defined by three main parameters: spreading factor (SF), bandwidth (BW), and carrier frequency [45].

SF can be summarized as the duration of the chirp. LoRa operates with SF from SF7 (the shortest time on air) to SF12 (the longest on air). Each level up in SF doubles the time on air to transmit the same amount of data. With the same bandwidth, longer time on air obviously results in less data transmitted per unit of time [9,46].

However, longer SFs are usually used when the EDs are far away from the GW. Although they reduce the data rate, they ensure a low probability of packet rejection. The SF configuration feature can be efficiently exploited by the GW in LoRaCog to ensure parallel functioning with LoRaWAN. When the GW sends the channel parameters using short SF, only the nearby EDs will be covered. Consequently, these EDs operate according to LoRaCog, while the faraway EDs will not be able to receive the GW’s message, and thus, they will remain performing the transmission as per LoRaWAN. Indeed, the value of the SF may be calibrated by the network to achieve the target number of the EDs to operate over the licensed spectrum.

#### 4.2.2. Bidirectional Communication

LoRaWAN grants bidirectional communication, where the system can communicate with EDs through receive windows. The NS can communicate with the EDs for better network performance and connectivity (such as time synchronization frame, channel parameters, etc.). The bidirectional connectivity makes the LoRaCog implementation simpler, since it allows the NS to send the channel parameterization to the EDs once the channel selection is done.

#### 4.2.3. Multi-Channel

LoRaWAN regional parameters can differ depending on the country, as defined in the LoRaWAN regional parameters document [47].

As an example, the European 868 MHz ISM frequency band defines a number of eight channels, which are each separated by a bandwidth of 300 KHz. Thus, if an ED wants to send a message, it fixes a spreading factor and the communication channel based on an NS instruction, and afterwards, it sends the message. The dynamism in channel allocation makes the CR usage for LoRa coherent with its fundamental configuration, since the available channels are dynamic.

As detailed above, several LoRaWAN features make CR adoption feasible. Indeed, these features should be adequately developed and modified to ensure the implementation of the CR.

#### 4.2.4. LoRa Gateway Role and Characteristics

GW, base station or access point are different terminologies for the same device that is responsible for radio exchange with LoRa devices and connected to NS by IP backhauls (fiber optics, cellular, WiFi, etc.).

GW receives LoRa-modulated RF messages from any ED at hearing distance and forwards these data messages to the NS. ED is not attached to any specific GW where it can be served by multiple gateways in the area. This arrangement significantly reduces the packet error rate (multiple receivers rather than one), reduces battery over consumption for mobile EDs, and opens the door for low-cost geo-location [44].

GWs operate entirely at the physical layer and only check the data integrity of each incoming LoRa RF message. If the integrity is not complete, the message will be dropped. In addition, they are used for downlinks; GW executes transmission requests coming from the NS without any interpretation of the payload.

GW’s main characteristics can be summarized as below:Main-powered most of the time, or solar where possible with backup batteries;Connected to external antennas (omni or directional);Installed in relatively high locations with the best line of sight;Have Ethernet or SimCard slots;Compact in design;Different types depending on the number of endpoints;Listening to eight different channels in parallel [48].

The GW features, such as the main power, the high locations, and the IP connectivity with the NS make it suitable for performing spectrum sensing. Being main-powered, the GW does not suffer power consumption problems when continuously performing spectrum sensing. The high location helps GW accurately harvest the licensed channel’s occupancy, since location may alleviate the channel conditions such as fading and shadowing. The IP connection ensures an existing support for reporting the spectrum sensing to the NS.

## 5. LoRaCog Protocol

LoRaCog is a new protocol using CR technique for IoT. As described earlier, it is born from LoRaWAN with changes applied on protocol’s operation and its components to successfully implement CR. The main advantage is to reduce traffic on the unlicensed spectrum whenever possible without impacting the quality of service (QoS) or the power consumption. Another main advantage is that LoRaCog will not replace LoRaWAN, but it can be considered as an assistant when the unlicensed spectrum becomes highly congested. Thus, a portion of the LoRa EDs will switch to the licensed spectrum to alleviate the congestion on the unlicensed spectrum. The sensing job is assigned to the GW to avoid adding an additional burden on the ED’s battery. The GWs are usually main powered, so there are no constraints on energy consumption. The single modification applied on the EDs will not need more energy due to the location change of the receive window only.

### 5.1. LoRaCog Mechanism

Maintaining the star topology of LoRaWAN, LoRaCog will operate in the same way. The EDs will send frames to NS through GWs in the range. As shown in Figure 4, the CR job will be split between GW for sensing, NS for deciding, NS and ED for acting. Hence, the EDs will not be doing any CR jobs except the transmission on the licensed channel selected by the NS in order to avoid any additional energy-consuming actions.

The GWs are doing spectrum sensing (it can be defined according to the country’s regulations) and feeding this information to the NS. The NS has pre-knowledge of EDs transmission time. This is ensured during the join-activation request. Modification will be done for the RX2 window, and prior to it, the NS will instruct the GWs to send the licensed spectrum holes with MAC command containing new channel’s parameters throughout their coverage of EDs. Spectrum holes in this case refer to the available unused licensed spectrum. Moving RX2 to just before TX (uplink transmission) is essential for the CR operation to inform the ED of the available channels to use. Using RX2, NS sends the available channel’s parameters to the ED through GW just before the transmission in order to maintain the relevance of the channel status, since the primary user may become active again at any time. Therefore, each time ED wants to transmit, a spectrum sensing operation should be done in order to provide it with the currently available channels.

The default channels defined in LoRaWAN will remain the same for each ED (for example, EU: three channels for uplink and one for RX2). They will be used for two purposes: the first is to receive any MAC command (such as the channel’s new configuration) since default channels are known and fixed for both GWs and EDs. Secondly, in case the ED does not receive the new configuration, it will continue operating on its uplink default assigned channels as per LoRaWAN regional parameters [47]. This will ensure successful transmission over the unlicensed ISM bands (for example, EU default channels).

LoRaCog will embed modification to the receive windows for classes A and B in LoRaWAN (see the next section); the NS will be able to send a synchronization frame (or new configuration frame) to inform and instruct the EDs (SUs in CR terminology) with available spectrum holes that the last can reconfigure and transmit accordingly.

Therefore, before each transmission, the EDs will open to receive a new configuration using RX2. Taking into consideration licensed user sudden appearance; the uplink (TX) will occur straightforwardly after RX2 without delay. Otherwise, the ED will transmit using its default channels within the unlicensed spectrum. Figure 5 illustrates the logic by flow chart demonstration.

### 5.2. LoRaCog End-Device

On the ED side, LoRaCog imposes two modifications: the first one concerns the join/activation request, and the second is a new arrangement for the default receive window.
Activation/Join request:During the activation, each ED should share the transmission time with the NS. The NS will use this information to send available spectrum holes to ED prior to its transmission time. The ED will be addressed uniquely through GW with the best RSSI.Receive window:Unlike LoRaWAN, the receive window RX2 will be shifted to before the uplink with no delay in between. This window will be used to receive the MAC commands from NS which could be the channel’s parameter configuration based on the sensing identified by the GW. The motivation behind this shifting is related to the relevance of the sensing decision. That means the available channels diagnosed by the GW and selected by the NS to use are not guaranteed to be free for a long time, since PU may become active at any time. Thus, the EDs should be informed of the channel to use just before transmitting. As shown in Figure 6, the uplink for ED will start right after the RX2. The RX1 will remain at a scheduled time after each transmission.

The purpose for choosing RX2 and not RX1 is that in LoRaWAN, RX2 uses a predefined channel and a fixed data rate known to all EDs while RX1 uses the channel of the last Uplink [42]. The default parameters in EU are 869.525 MHz/DR0 (SF12, 125 kHz) [47]. Hence, the gateway can manage the EDs using such controlled channels for configurations. In this case, the channel decision, such as the SF, the channel frequency and the data rate will be communicated to the ED just before the uplink transmission.

Then, the ED wakes up, receives the frame and decides to transmit directly or configure before transmitting. It is important to mention that the ED, in case of no instructions from NS, will switch back to default channels and not use the ones used in the previous uplink.

### 5.3. LoRaCog Gateway

The GW will have CR capability; it will scan the spectrum continuously for holes and identify the possible channels to be used in the licensed spectrum. In addition, it is able to communicate on different spectrum channels supported by required hardware. The spectrum sensing is assigned to the GWs and not to the EDs, since the former are usually main-powered. Thus, there is no energy constraint related to this additional operation. Moreover, the GWs are installed at high altitude, providing them with good communication conditions with PUs. This kind of installation is not usually ensured for the EDs.

The sensing results will be forwarded from the GWs to the NS, where the latter will decide which channels to be forwarded to EDs and when to match the RX2 window, respectively. Each GW will receive its related instructions to be forwarded to the EDs.

The transmission from GW should use a default channel which is known and fixed for all EDs. Since the purpose of this protocol is to reduce unlicensed spectrum congestion, the transmission will be done using the lowest SF (SF7) to reduce collision risk with uplinks from EDs. The decision to use the lowest SF will achieve better efficiency for spectrum and ensure the parallel functioning of both LoRaCog and LoRaWAN due to the below:Far EDs will most likely not receive the frame, and hence, they will operate normally using the normal LoRaWAN mechanism; that is, ED can choose channels allocated and permitted by regional parameters. This will reduce the possibility of collision and ensure fair spectrum usage.The nearby EDs will switch to available licensed (cognitive) spectrum to use the holes indicated by GW as they are in the range. The time over the air (OTA) will be less in this case, which is a safety measure not to interrupt any PU that might show up.

### 5.4. LoRaCog Network Server

The change for NS will impose a decision task for the CR sensing done by GW to decide on the best available channel. The NS using the obtained database for EDs transmission time will look for the best unused holes for each GW, in case they exist, and forward to ED uniquely through the MAC commend prior to ED transmission. The role of NS in deciding the best conditioned channels may be based on the history and prediction intelligence [49,50], where channels with a high probability of PU re-appearance should be avoided to prevent the collision between the two.

As NS deletes duplicated frames coming from the same ED through different GWs and keeps the one with the best RSSI in LoRaWAN, NS will apply the same mechanism to select the best GW to address each ED in LoRaCog.

The NS will include a network session key (NwkSKey); this will avoid EDs receiving contradicting instructions from different GWs in their ranges. Hence, the GW that has the best RSSI from EDs through their uplinks, will be the one used to send the spectrum holes, and EDs will not listen to other GWs in the range.

To maintain consistency between EDs time and NS time, LoRaCog will be configured to send synchronization frame along with the frame acknowledgment after receiving the EDs data. This frame will be received by EDs during their RX1, which is sent with a delay to match RX1’s opening time.

### 5.5. LoRaCog Architecture

As it is clear now, the objective is to reduce the load on an unlicensed spectrum. Many EDs will be given the opportunity to use the licensed spectrum, and this will create better spectrum efficiency.

Summarizing previous sections for LoRaCog, below is a simple illustration for the architecture mainly around the new mechanism. It shows the main steps in the protocol legend by the numbers in Figure 7 as follows:1.Each GW is sensing spectrum for holes;2.Then, the GW sends sensing results to NS;3.NS decides on the best available unused spectrum for each ED to avoid PU in frequently used channels;4.Each GW forwards the available spectrum from NS to its EDs;5.EDs receive (or not) the available spectrum and transmit accordingly.

## 6. Performance Evaluation and Simulation Environment

Demonstrating the functionality and benefits of LoRaCog, simulations are conducted to evaluate the protocol against the rejected data packets compared to LoRaWAN. This section will describe the method used in the simulation, the comparison and the results.

The communication parameters between EDs and GWs are the same for LoRaCog as in LoRaWAN (such as SF and sensitivity), with the only difference for LoRaCog being a wider frequency range to include the licensed spectrum.

The main driver for the LoRaCog protocol is to ensure spectrum efficiency by utilizing the available unused frequencies (licensed or unlicensed) and reduce the congestion on unlicensed bands. The congestion will normally result in high collision or rejected packets when EDs are trying to communicate in non-cooperative manners.

The rejected packet rate is a function of the channel’s sensitivity between the EDs and GWs. This means that any sent packet with RSSI below the sensitivity level will not be detected by the GW and considered as rejected. The simulation is based on the known and used Okumura-Hata’s model [51] for radio frequency communication systems in urban areas to calculate the path loss and link budget. This model considers radio propagation in different areas as well as the impact of city buildings and environments on diffraction, reflection and scattering [52]. Ref. [53] relied on Okumura-Hata to estimate path loss in urban areas for UHF (Ultra High Frequency) and VHF (Very High Frequency) land mobile radio services. The objective is to calculate the received power at the GW from each ED:(1)$$P_{RX}=P_{TX}+G_{TX}-L_{TX}-L_{FS}-L_{M}+G_{RX}-L_{RX}$$
where
$P_{RX}$: Received Signal Power (dBm);$P_{TX}$: Transmitter Signal Power (dBm);$G_{TX}$: Transmitter Antenna Gain (dBi);$L_{TX}$: Transmitter Losses (dBm);$L_{FS}$: Free Space Loss or Path Loss (dBm);$L_{M}$: Miscellaneous Losses (dBm);$G_{RX}$: Receiver Antenna Gain (dBi);$L_{RX}$: Receiver Losses (connectors) (dBm).

### Simulation Methodology

The simulation is built using real site estimation where LoRa networks are deployed. The network is assumed to cover an urban area of a 2000 m radius using 4 GWs and having 500 EDs. The GWs and EDs are uniformly distributed in the area with one GW located in the middle as represented in Figure 8.

Since LoRaWAN allows multiple channels depending on regional parameters, ED’s number was chosen to be 500 to represent a realistic quantity that can be found communicating on one single channel (same for other channels). The simulation assumed that each ED would send one packet in the air, and the GWs, depending on the RSSI level compared to their sensitivity threshold, would reject or decode the message. Unless otherwise specified, the configuration and parameters used in the simulation are described in Table 1. These parameters are used to calculate path-loss and rejected packets based on LoRaWAN regional parameters [54,55] and defined in Matlab as a matrix of packet duration, GW sensitivity, and SF (signal to interference and noise ratio). This is in addition to other parameters for radio from Okumara-Hata such as the antenna’s height and gain, power, and the ED’s antenna height. The NS will identify most free bands and allocate them to EDs accordingly. For simplicity, we have used two frequencies in LoRaCog, the original EU 868 MHz from LoRaWAN and an additional licensed frequency 438 MHz [56]. The simulation took into consideration different split ratios between 868 and 438 MHz, as illustrated later. For example, LoRaCog 80 by 20 would mean 80% of EDs are operating on 868 MHz and 20% are operating on 438 MHz.

## 7. Results

The evaluation is based on comparison of the rejected packet rate between LoRaWAN and LoRaCog using different scenarios that will be explained in this section.

As a reminder and for demonstration purposes, the simulation is using 868 MHz frequency for LoRaWAN (such as in EU) and an additional cognitive frequency 438 MHz for LoRaCog along with 868 MHz. LoRaCog can be configured to utilize multiple licensed or unlicensed frequencies, which will generate better performance.

Figure 9 illustrates the comparison between LoRaWAN 868 MHz and different splits of LoRaCog EDs between 868/438 MHz bands. For both protocols, 4 GWs were used to detect the packets from 500 EDs. If a packet is received by any of the 4 GWs, it will be successfully decoded and not rejected. The result is better for LoRaCog, and rejected packets decrease when the EDs are more distributed among the cognitive frequencies. In this example, LoRaWAN with 500 EDs, the rejected packets percentage was above 50% while using LoRaCog, it goes down to 24% (using 10 by 90 split).

To illustrate the effect of the proposed LoRaCog on the number of needed GWs, Figure 10 compares LoRaWAN when using 4 and 6 GWs with LoRaCog using 4 GWs. The figure shows that LoRaWAN with 6 GWs has the same performance as LoRaCog 4 GWs when 60% of EDs are on 868 MHz and the rest are on 438 MHz. However, LoRaCog has lower rejected packets from LoRaWAN 6 GWs when the percentage of the EDs operating on the cognitive channel (438 MHz) increases. Therefore, LoRaCog can reduce the investment of infrastructure, and at the same time, it assures spectral efficiency that is utilized fairly between EDs, avoiding congestion where possible.

The forthcoming simulation considers the probability of the re-appearance of PUs. PU may return to be active on the cognitive channel, leading to a collision with ED transmission. However, LoRaCog may strongly reduce the effect of this re-appearance for two main reasons. Primarily, the NS will be doing analysis for the spectrum sensed by GWs on a continuous basis and will recommend only those with very low activities from PUs. Secondly, LoRaCog protocol is acting similarly to LoRaWAN and its characteristics; thus, ED will be sending short messages in milliseconds and not frequently (once per day for battery operated EDs), so it will not interfere with PUs.

Figure 11 shows the performance of LoRaCog compared to LoRaWAN assuming PUs will come back to the network with a probability of 1%. PUs interrupt the transmission of EDs using cognitive frequency. The active number of PUs are simulated as an equal number of rejected packets from those EDs on the cognitive side. LoRaCog shows better performance even with active PUs in the network and in parallel reduces congestion.

Figure 12 illustrates the behavior of LoRaCog compared to LoRaWAN for 5% probability of PUs re-appearance on the operating cognitive frequencies. LoRaCog still performs better than LoraWAN for the different split combinations for both scenarios: when the majority of EDs operate on the cognitive channels (Figure 12a) or when operating on the classical channels (Figure 12b). However, it is clear that the PU re-appearance probability affects performance when comparing the results of Figure 12 to those of Figure 11. It is worth mentioning that the split 50by50 is the more reliable combination, since it presents the less rejected packets.

In order to show the behavior of the proposed LoRaCog for high re-appearance PU probability, Figure 13 shows the performance of LoRaCog with 10% probability of PUs’ re-appearance in cognitive channels. The cognitive usage will not be useful in this case. This scenario highlights the role of NS in selecting the most reliable channels to allocate to EDs. It is insufficient to allocate a free channel to EDs, but the probability of re-appearance of the PU on this channel is also an important factor. Here, NS may exploit the historical behavior of the PU to make a reliable decision on the allocated channels by selecting the least occupied channels [57,58].

## 8. Conclusions

In this paper, we proposed a new protocol for Cognitive Radio-based LoRa network, which is called LoRaCog. LoRaCog ensures the fair usage of spectrum and maintains suitability for battery-operated devices. Furthermore, it enhances the communication between EDs and GWs accordingly. In fact, it is a complementary protocol for LoRaWAN used when the spectrum is congested or limited.

LoRaCog is inspired by LoRaWAN by implementing CR to manage spectrum sensing and available channel decisions. This will be accompanied with changes to the ED RX2 receive window. The GWs should have modifications in hardware and software to empower spectrum sensing as suggested earlier and the NS to embed the capability to authorize the usage of the least used channels after receiving the sensing results from the GWs. These modifications will have no impact on battery consumption for EDs, since sensing is done by GW and hence defeats the main challenge of CR in high power consumption.

Then, we configured our system model to compare LoRaWAN and LoRaCog in different scenarios. The results show the efficiency of LoRaCog compared to LoRaWAN. It improves the QoS and decreases collisions. It also allows us to decrease the number of GWs and thus the network cost. For future work, we will be looking at the GWs hardware to manage spectrum sensing over different bands and dynamically move within the spectrum. In addition, machine learning techniques are added to the NS for deciding on the best available channels in a short time by looking at historical activities of primary users and the EDs’ expected transmission time to avoid any collision or service disturbance. 

## Figures and Tables

**Figure 1 sensors-22-03885-f001:**
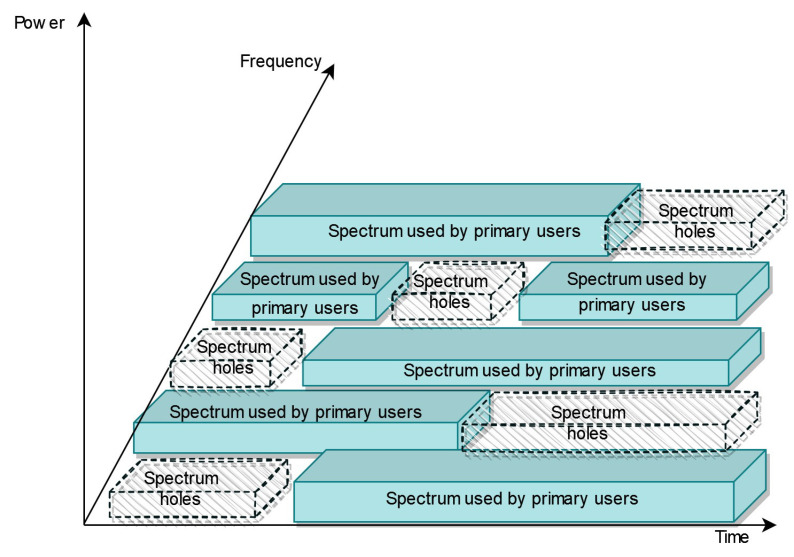
Spectrum holes: The CR user is able to access the available channels and is obliged to avoid the occupied ones in order to minimize the interference [27].

**Figure 2 sensors-22-03885-f002:**
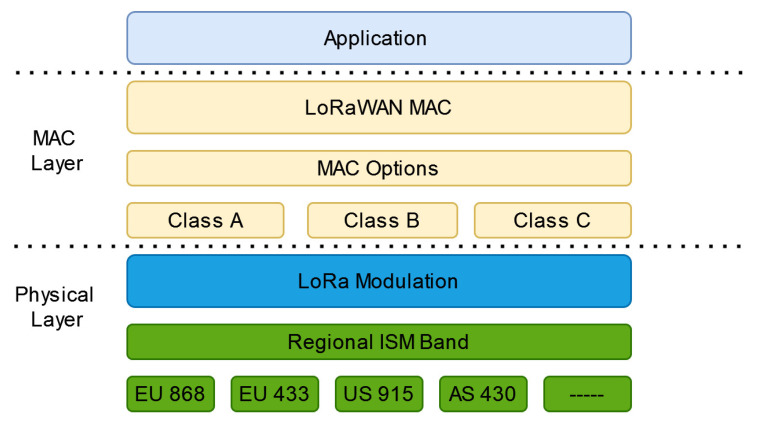
LoRaWANProtocol Stack [44].

**Figure 3 sensors-22-03885-f003:**
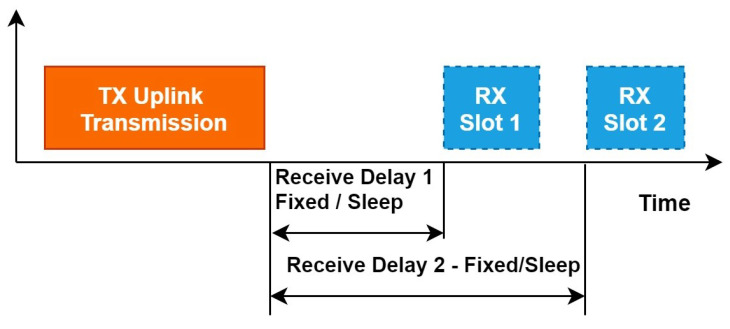
Class A operation.

**Figure 4 sensors-22-03885-f004:**
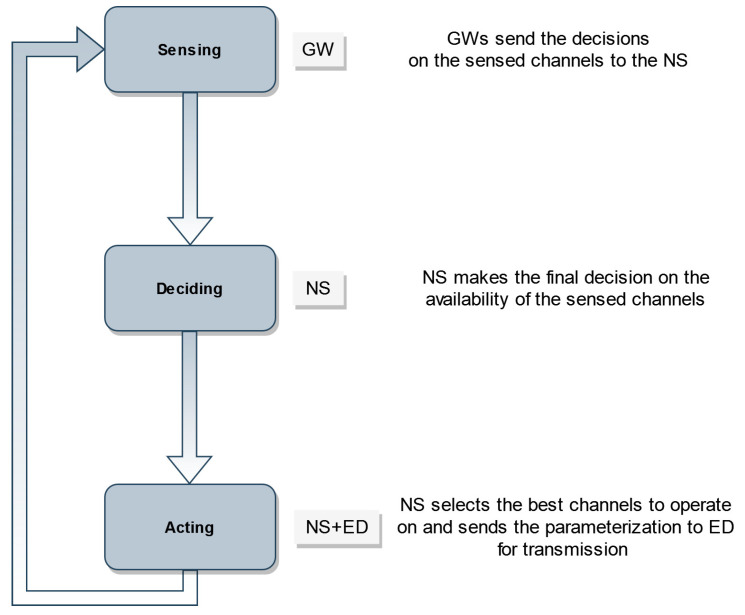
Cognitive Radio Cycle managed by LoRaCog.

**Figure 5 sensors-22-03885-f005:**
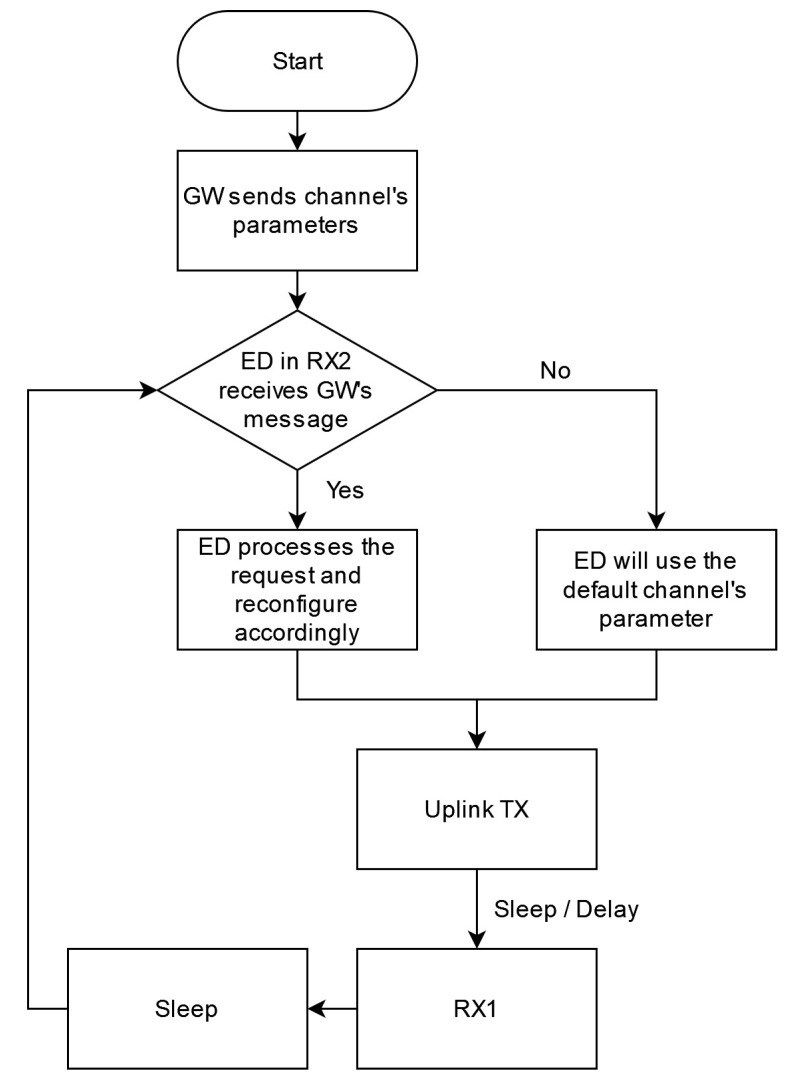
LoRaCog Logarithm.

**Figure 6 sensors-22-03885-f006:**
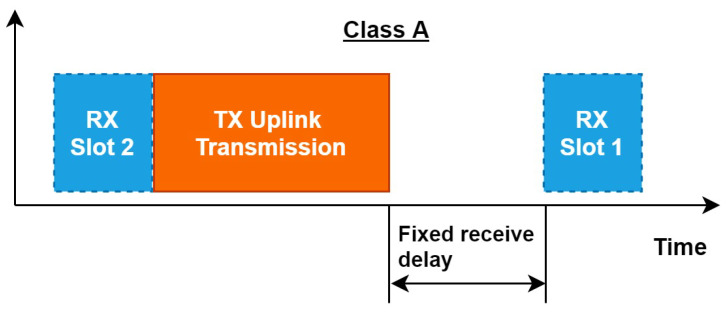
LoRaCog Class A Operation.

**Figure 7 sensors-22-03885-f007:**
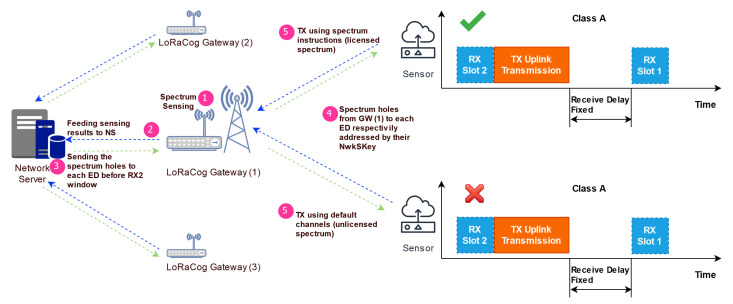
LoRaCog Global Architecture.

**Figure 8 sensors-22-03885-f008:**
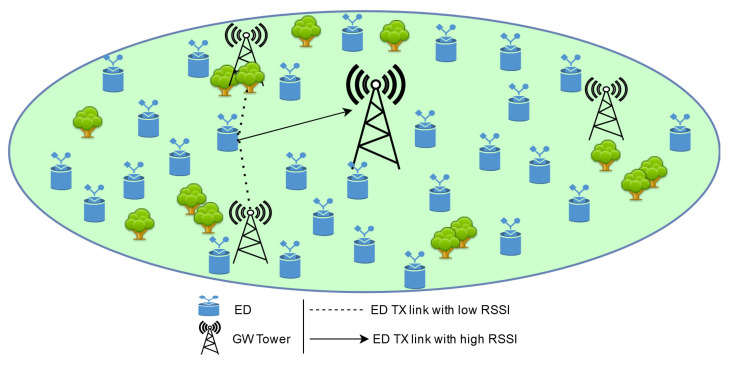
Representation of the EDs and the GWs distribution in our simulations.

**Figure 9 sensors-22-03885-f009:**
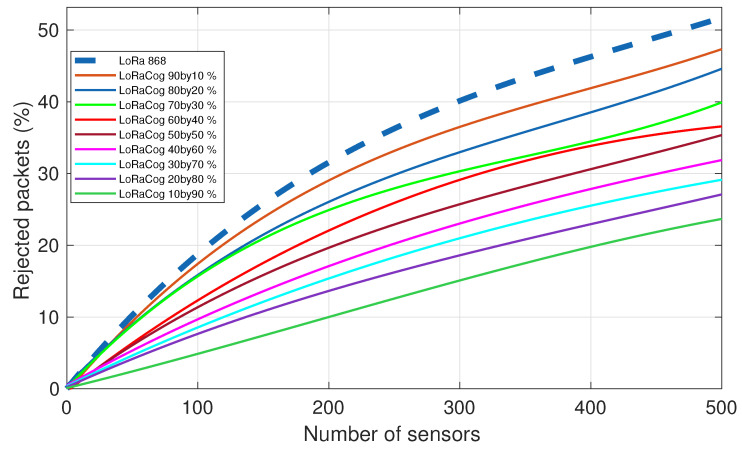
The evolution of the rejected packets for both LoRaWAN 868 MHz and LoRaCog 868/438 MHz for different percentage splits.

**Figure 10 sensors-22-03885-f010:**
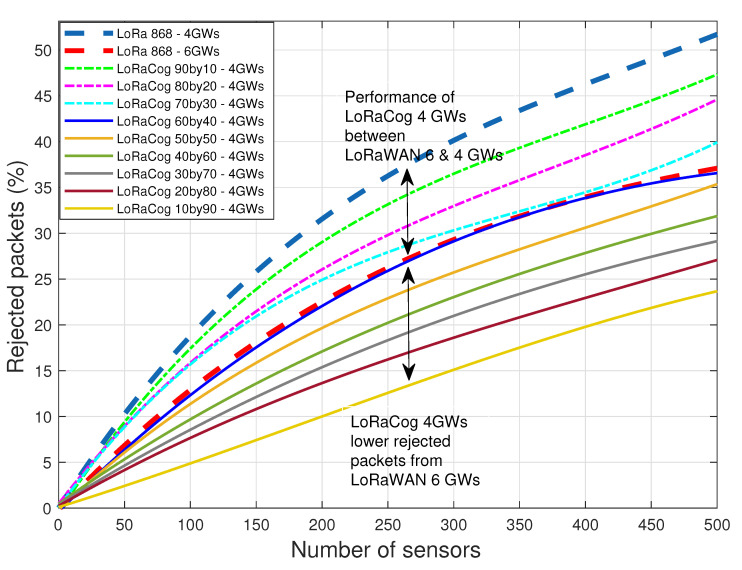
LoRaWAN 4 and 6 GWs vs. LoRaCog 4 GWs.

**Figure 11 sensors-22-03885-f011:**
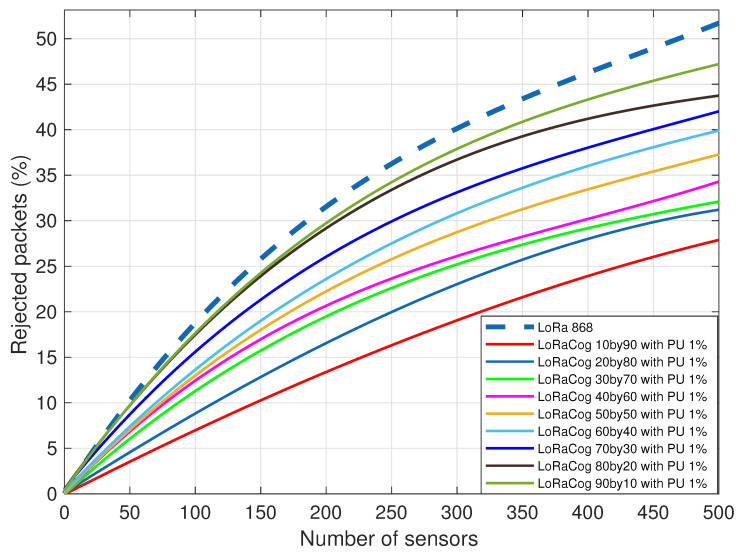
LoRaWAN 868 MHz vs. LoRaCog 868/438 MHz with PU 1%.

**Figure 12 sensors-22-03885-f012:**
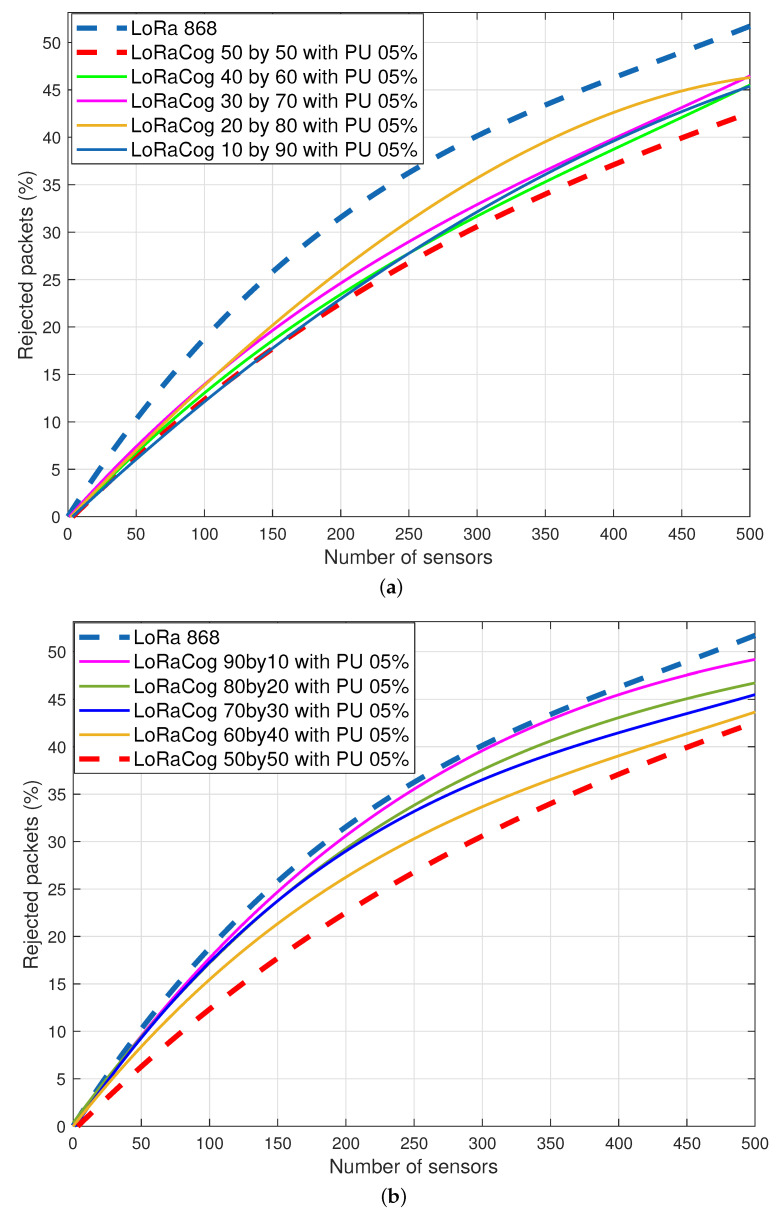
Evolution of the rejected packets for LoRaWAN and LoRaCog for a probability of re-appearance of 5%. (**a**) More than 50% of the EDs are operating on the cognitive channel. (**b**) Less than 50% of the EDs are operating on the cognitive channel.

**Figure 13 sensors-22-03885-f013:**
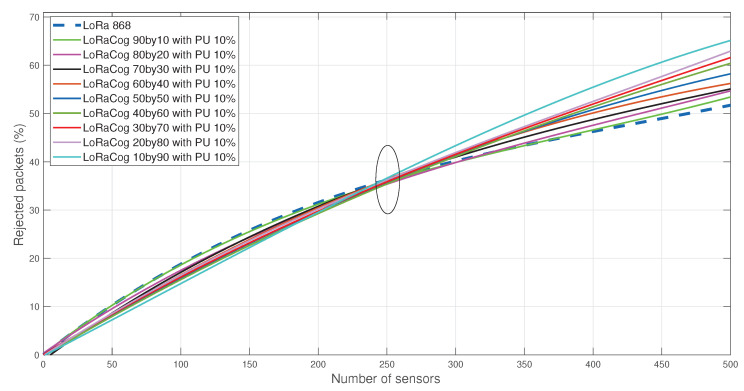
LoRaWAN 868 MHz vs. LoRaCog 868/438 MHz with PU re-appearance probability = 10%.

**Table 1 sensors-22-03885-t001:** Matlab Simulations Parameters.

Parameter	Details
Number of GWs	4
Number of EDs	500
Distribution of ED and GWs	Uniformly distributed
Frequency	LoRa 868 MHz/LoRaCog 868 & 438 MHz
Power	25 mW
Bandwidth	125 KHz
Area	Small–medium city
Simulation	Monte Carlo
Model	Okumura-Hata
Spreading Factors	Same as LoRaWAN
Sensitivity	Same LoRaWAN

## Data Availability

Not applicable.

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
