# Peer review of "LoRaCog: A Protocol for Cognitive Radio-Based LoRa Network"

_sensors, 2022, doi:10.3390/s22103885_

Round 1

Reviewer 1 Report

The paper proposes a new scheme by adding CR to LoRaWAN to reduce the congestion and maintain Lo-RaWAN’s suitability for IoT devices. The paper can be enhanced by addressing the following:

1- The abstract needs revision to state clearly what the paper is proposing with some significant results. 
2- The introduction is missing the contribution part of the research study. Authors need to add this part before the last paragraph in the introduction to highlight what is the contribution of the paper. This can be presented as a few bullet points.
3- Authors need to change the sections evaluation and simulation into, performance evaluation and simulation environment and results should be presented in a separate section. 
4- All acronyms should be given in a table.

5- The paper needs proofreading to fix the errors.

6- The authors are encouraged to explain/evaluate the performance of the proposed scheme under channel fading conditions.

7- How do explain what would the performance of the proposed scheme under interference conditions.

Author Response

First of all, we would like to thank the editor and the unknown reviewers for their constructive and useful comments and feedback. Hereinafter, we briefly emphasize and discuss how we have addressed the various points raised by the reviewers. Please note that the corrections in the new version of the paper are written in blue. Our responses below to the reviewer’s comments are in bold.

Further details are given as follows:

Reviewer 1:

Comments to the Author

The paper proposes a new scheme by adding CR to LoRaWAN to reduce the congestion and maintain LoRaWAN’s suitability for IoT devices. The paper can be enhanced by addressing the following:

  1. The abstract needs revision to state clearly what the paper is proposing with some significant results

The abstract is revised accordingly and highlighted in the manuscript in blue color.

  1. The introduction is missing the contribution part of the research study. Authors need to add this part before the last paragraph in the introduction to highlight what is the contribution of the paper. This can be presented as a few bullets points.

The introduction is revised to include the contribution part.

  1. Authors need to change the sections evaluation and simulation into, performance evaluation and simulation environment and results should be presented in a separate section.

The former section evaluation and simulation is changed into two sections, section 6 called performance evaluation and simulation environment and section 7 named results.

  1. All acronyms should be given in a table

The Acronyms table is added to the introduction section:

LoRaCog

The new proposed protocol

IoT

Internet of Things

LPWAN

Low Power Wide Area Network

LoRaWAN

LoRa LPWAN protocol

CR

Cognitive Radio

ISM

Industrial, Scientific and Medical

ED

End Devices

RX

Receive Window

Tx

Uplink transmission

GW

Gateway

NS

Network Server

RSSI

Radio Signal Strength Indicator

AI

Artificial intelligence

SDR

Software Defined Radio

CSS

Chirp-Spread Spectrum

SF

Spreading Factor

QoS

Quality Of Service

TV

Television

MAC

Medium Access Control

EU

European Union

BW

Bandwidth

PU

Primary User

SU

Secondary User

OTA

Over The Air

NwkSKey

Network session key

  1. The paper needs proofreading to fix the errors

The proof reading was performed, and mistakes were corrected.

  1. The authors are encouraged to explain/evaluate the performance of the proposed scheme under channel fading conditions.

In our simulations we investigated the performance of our protocol under realistic channels conditions. For this reason, we adopted the Okumura-Hata model on channel fading that is adopted by many research works. To clarify this issue, we added the following clarification to the Section 6:

“This model considers radio propagation in different areas, impact of city buildings and environments on diffraction, reflection, and scattering [54]. [ 55 ] relied on Okumura Hata to estimate path loss in urban areas for UHF (Ultra High Frequency) and VHF (Very High Frequency) land mobile radio services.”

  1. How do explain what would the performance of the proposed scheme under interference conditions

The interference can be from different sources. The paper considered the interference between EDs each other and/or with primary users when they become active. To eliminate the interference factor, LoRaCog is flexible enough to distribute EDs into the many available channels in a coherent way. Moreover, the protocol is using the lowest SF which achieve better efficiency for spectrum and ensure the parallel functioning (i.e., less interference) of both LoRaCog and LoRaWAN due to the below:

  • Far EDs will most likely not receive the frame and hence they will operate normally using the normal LoRaWAN mechanism, that is, ED can choose channels allocated and permitted by regional parameters. This will reduce possibility of collision and ensure fair spectrum usage.
  • The near-by EDs will switch to available licensed (cognitive) spectrum to use the holes indicated by GW as they are in the range. The time over the air (OTA) will be less in this case, which is a safety measure to not interrupt any PU that might show up.

This issue is highlighted in the section 5.3 “LoRaCog Gateway” of the paper.

Reviewer 2 Report

Overall, the paper is well-written and the proposed extension to the LoRaWAN protocol, namely LoRaCog, had been presented quite clearly in the manuscript. The merits and performance of the proposed protocol extension had also been demonstrated suitably in Section 6. In my opinion, the results would certainly be of significant interest to those that are seeking to improve the performance of LoRaWAN operations.

However, my one slight criticism in the proposed protocol is in the handling of the spectrum sensing operation by the GW on behalf of the EDs. While the merits of shifting this operation from the ED to the GW has been identified in the manuscript, one may question if it can indeed perform adequately on behalf of the ED, given that both the GW and ED may be quite far apart and may not sufficiently capture the correct spectrum conditions around that ED. I suspect that this will be further investigated in future work.

Author Response

First of all, we would like to thank the editor and the unknown reviewers for their constructive and useful comments and feedback. Hereinafter, we briefly emphasize and discuss how we have addressed the various points raised by the reviewers. Please note that the corrections in the new version of the paper are written in blue. Our responses below to the reviewer’s comments are in bold.

Further details are given as follows:

Reviewer 2:

Comments to the Author

Overall, the paper is well-written and the proposed extension to the LoRaWAN protocol, namely, LoRaCog, had been presented quite clearly in the manuscript. The merits and performance of the proposed protocol extension had also been demonstrated suitably in Section 6. In my opinion, the results would certainly be of significant interest to those that are seeking to improve the performance of LoRaWAN operations.

However, my one slight criticism in the proposed protocol is in the handling of the spectrum sensing operation by the GW on behalf of the Eds. While the merits of shifting this operation from the ED to the GW has been identified in the manuscript, one may question if it can indeed perform adequately on behalf of the ED, given that both GW and ED may be quite far apart and may not sufficiently capture the correct spectrum conditions around that ED I suspect that this will be further investigated in the future work.

This can be investigated further in the future work. However, the GW is expected to give the same accuracy as if the ED is doing the sensing for below reasons:

  • GW is installed at relatively high places with good open sight which ensures coverage and connectivity with ED and hence accurate sensing
  • Multiple GWs will be sensing the same area most of the time, so capturing entire spectrum around the area is ensured and the NS will identify the best available channel with lowest chance of primary user’s activities.
  • Since we are keeping the regional spectrum for LoRaWAN and adding to it the available unused spectrum as per LoRaCog, the NS, and based on RSSI, will send the desired channels through GWs over lowest SF to avoid far ED from sending on cognitive channel which will take longer time over the air and hence more probability for collision.

Reviewer 3 Report

While the work is interesting, I have the following concerns:

  1. How was the LoRa signal created in matlab? Was it only based on CSS modulation only?
  2. Can the authors add an energy study? While their method is clearly better in terms of packets received, it obviously will consume more energy
  3. The authors do not dicuss what those holes' statistics or values are. Other studies have worked on cognitive radio for IoT applications:
    1. https://ieeexplore.ieee.org/document/9205874
    2. https://ieeexplore.ieee.org/document/9402690

Author Response

First of all, we would like to thank the editor and the unknown reviewers for their constructive and useful comments and feedback. Hereinafter, we briefly emphasize and discuss how we have addressed the various points raised by the reviewers. Please note that the corrections in the new version of the paper are written in blue. Our responses below to the reviewer’s comments are in bold.

Further details are given as follows:

Reviewer 3:

Comments to the Author

While the work is interesting, I have the following concerns:

  • How was the LoRa signal created in matlab? Was it only based on CSS modulation only?

For the interest of our research, we are focused on the rejected packets and the received power. We have followed LoRaWAN parameters (frequency, power, SF, sensitivity, etc) as given in [56,57] to model the power received at the GW from each ED by calculating path loss and link budget. If the packet is with RSSI below the sensitivity level, it will not be detected by the GW and considered as rejected. We added the following paragraph to the Section 6.1. “Simulation Methodology” to the new version of the paper to highlight this issue:

 “These parameters are used to calculate path-loss and rejected packets based on LoRaWAN regional parameters [ 56, 57 ] and defined in Matlab as a matrix of packet duration, GW sensitivity, SF, Signal to interference and noise ratio. This in addition to other parameters for radio from Okumara-Hata like antenna’s height and gain, power, ED’s antenna height”

  • Can the authors add an energy study? While their method is clearly better in terms of packets received, it obviously will consume more energy.

The sensing job is assigned to the GW to avoid adding additional burden on ED’s battery. The GWs are usually main powered so no constraints on energy consumption. The single modification applied on the ED’s will not need more energy due to location change of the receive window only. This point is highlighted in section 5 “LoRaCog Protocol” of the new version of the paper.

  • The authors do not discuss what those hole’s statistics or values are. Other studies have worked on cognitive radio for IoT applications
    1. https://ieeexplore.ieee.org/document/9205874
    2. https://ieeexplore.ieee.org/document/9402690

Thanks for these two papers. We added them as references.

Round 2

Reviewer 1 Report

The authors have addressed all previous comments. Just minor spell check is encouraged.

Reviewer 3 Report

The paper is in better shape, however the English requires significant enhancement.